# CD131 contributes to ulcerative colitis pathogenesis by promoting macrophage infiltration

**Zhiyuan Wu[1†], Lindi Liu[2†], Chenchen He[1†], Lin Xiao[3], Duo Yun[4], Junliang Chen[1], Zhihao Liu[1], Wenjun Li[1], Qingjie Lv[1]\*, Xiaodong Tan[1]\***

[1]Department of General Surgery, Shengjing Hospital of China Medical University, Shenyang, China; [2]Department of Pathology, Shengjing Hospital of China Medical University, Shenyang, China; [3]Department of Gastroenterology, Shengjing Hospital of China Medical University, Shenyang, China; [4]Department of Oncology, Capital Medical University Affiliated Beijing Friendship Hospital, Beijing, China

**\*For correspondence:**
lvqjie@163.com (QL);
tanxdcmu@163.com (XT)

[†]These authors contributed equally to this work

**Competing interest:** The authors declare that no competing interests exist.

## eLife Assessment

Ulcerative colitis (UC) is a chronic gut inflammatory condition affecting the colon in humans. This study uses human samples as well as a mouse model of colitis induced by a chemical, DSS, to investigate the role of an immune marker, CD131, in UC pathogenesis. The study, as presented, is **incomplete**, as experimental details are lacking, the statistical analyses are deficient, and there is not yet direct evidence for a CD131-mediated mechanism of gut inflammation.

## Abstract

**Background:** Ulcerative colitis (UC) is a group of chronic inflammatory bowel disease mainly affecting the colon. The exact etiology of UC remains elusive. CD131 is a receptor subunit mediating the effects of hematopoietic growth factors granulocyte–macrophage colony-stimulating factor (GM-CSF) and Interleukin-3 (IL-3), which regulate various inflammatory responses. The pleiotropic effects of the cytokines on intestinal inflammation suggest that additional factors influence their overall function, where the receptor may play a role.
**Methods:** In the present study, we investigated the role of CD131 in the pathogenesis of UC, with the use of murine colitis model established by administration of dextran sulfate sodium (DSS) in the drinking water.
**Results:** By comparing the immune and inflammatory responses between wt and CD131-deficient mice, we found that CD131 contributed to DSS-induced murine colitis, which functioned in synergy with tissue-infiltrating macrophages. Besides, CD131 may have promoted the chemotaxis of macrophages and T cells into the colon through CCL4. In addition, we analyzed clinical data and pathology specimens from UC patients and found that CD131 was associated with the endoscopic and pathological severity of intestinal inflammation.
**Conclusions:** The present study provides a novel way to the understanding of the mechanisms of GM-CSF and IL-3 effects in the intestine, which will benefit the development of therapeutic approaches.
**Funding:** The present work has received no external funding but only from the affiliated institution.

## Introduction

Ulcerative colitis (UC) is a group of chronic inflammatory bowel disease (IBD) mainly affecting the colon (*Ungaro et al., 2017*). The exact etiology of UC remains elusive. Available evidence suggests that IBDs are caused by dysregulated immune responses against components of the microbiota and intestinal wall in genetically susceptible individuals (*Xavier and Podolsky, 2007*; *Chang, 2020*). The adaptive immunity is classically considered to play a major role in the pathogenesis of IBDs; however, accumulating evidence has shown that the innate immunity is equally as important (*Geremia et al., 2014*). IBDs may result from impaired functioning of the intestinal innate immune system, which is comprised of the epithelial barrier and a variety of innate immune cells, including neutrophils, monocytes/macrophages, and dendritic cells (*Parikh et al., 2019*; *Drury et al., 2021*; *Na et al., 2019*; *Shale and Ghosh, 2009*).

Granulocyte–macrophage colony-stimulating factor (GM-CSF) and Interleukin-3 (IL-3) are a group of hematopoietic growth factors for the myeloid lineage mainly functioning under inflammatory conditions (*Becher et al., 2016*; *Dougan et al., 2019*). They regulate various inflammatory responses that promote rapid clearance of pathogens while contribute to the persistence of chronic inflammation (*Dougan et al., 2019*). Besides, these cytokines also mediate the cross-talk between innate and adaptive immunity (*Dougan et al., 2019*). The receptors for GM-CSF and IL-3, expressed on various immune cells, are both heteromers comprising a ligand-specific α subunit (CD116 and CD123, respectively) and a shared β common subunit (CD131) (*Becher et al., 2016*; *Dougan et al., 2019*). The α chains provide specificity for the cytokines but bind with relatively low affinity, while their association with the β common chain provides high affinity necessary for effective downstream signaling (*Broughton et al., 2012*; *Hansen et al., 2008*).

The role of these cytokines in the pathogenesis of IBDs has been investigated but is still controversial. For instance, GM-CSF has been shown to deliver pleiotropic effects, either pro- or anti-inflammatory, on the intestinal inflammation depending on the cellular and cytokine milieu as well as disease model (*Castro-Dopico et al., 2020*; *Xu et al., 2008*). The different results imply that additional factors influence the overall function of GM-CSF, including the expression and functional status of its receptor. For example, a previous study showed that IBD was characterized by impaired expression and function of CD116 in circulating granulocytes and monocytes, suggesting a defect in innate immunity in response to GM-CSF (*Goldstein et al., 2011*). Besides, in a genetic analysis of Ashkenazi Jewish populations that have a high prevalence of Crohn's disease, a frameshift mutation in CSF2RB gene (encoding CD131) was shown associated with higher risk for Crohn's disease and reduced monocyte signaling in response to GM-CSF (*Chuang et al., 2016*). Indeed, a better understanding of the effect of GM-CSF receptor in the pathogenesis of IBD is necessary.

Here, we investigated the role of CD131, the β common chain mediating the effects of GM-CSF and IL-3 signaling, in the pathogenesis of UC. We established murine colitis model by administration of dextran sulfate sodium (DSS) in drinking water and compared the immune and inflammatory responses between wild-type and CD131-deficient mice. Besides, we analyzed clinical data and pathology specimens from UC patients to unravel its clinical significance.

## Results

### CD131 contributed to DSS-induced murine colitis

We firstly explored the change of *Csf2rb* gene (encoding CD131) expression in murine colon tissues in response to DSS administration in the drinking water by analyzing the gene expression dataset GSE22307 (*Fang et al., 2011*). The expression of *Csf2rb* appeared to elevate with time, but only until day 6 a significantly increased *Csf2rb* expression was demonstrated (*Figure 1a*). We then subjected our wt mice to 2% DSS in the drinking water for 7 days to establish murine colitis and found that *Csf2rb* mRNA expression was significantly elevated in the distal part of the colon comparing to mice receiving normal drinking water (*Figure 1b*). The *Csf2rb* mRNA could not be effectively detected in the proximal part of the colon and therefore only the distal colon was used for further analyses. Nevertheless, we could not find any significant change in *Csf2rb* mRNA expression level in other immune-related tissues, including blood leukocytes, intra-peritoneal lavage cells, mesenteric lymph nodes, spleen, and thymus, following DSS administration (*Figure 1—figure supplement 1a–e*). The gene expression dataset analysis also showed a tendency of increase in *Csf2ra* (*Figure 1—figure*

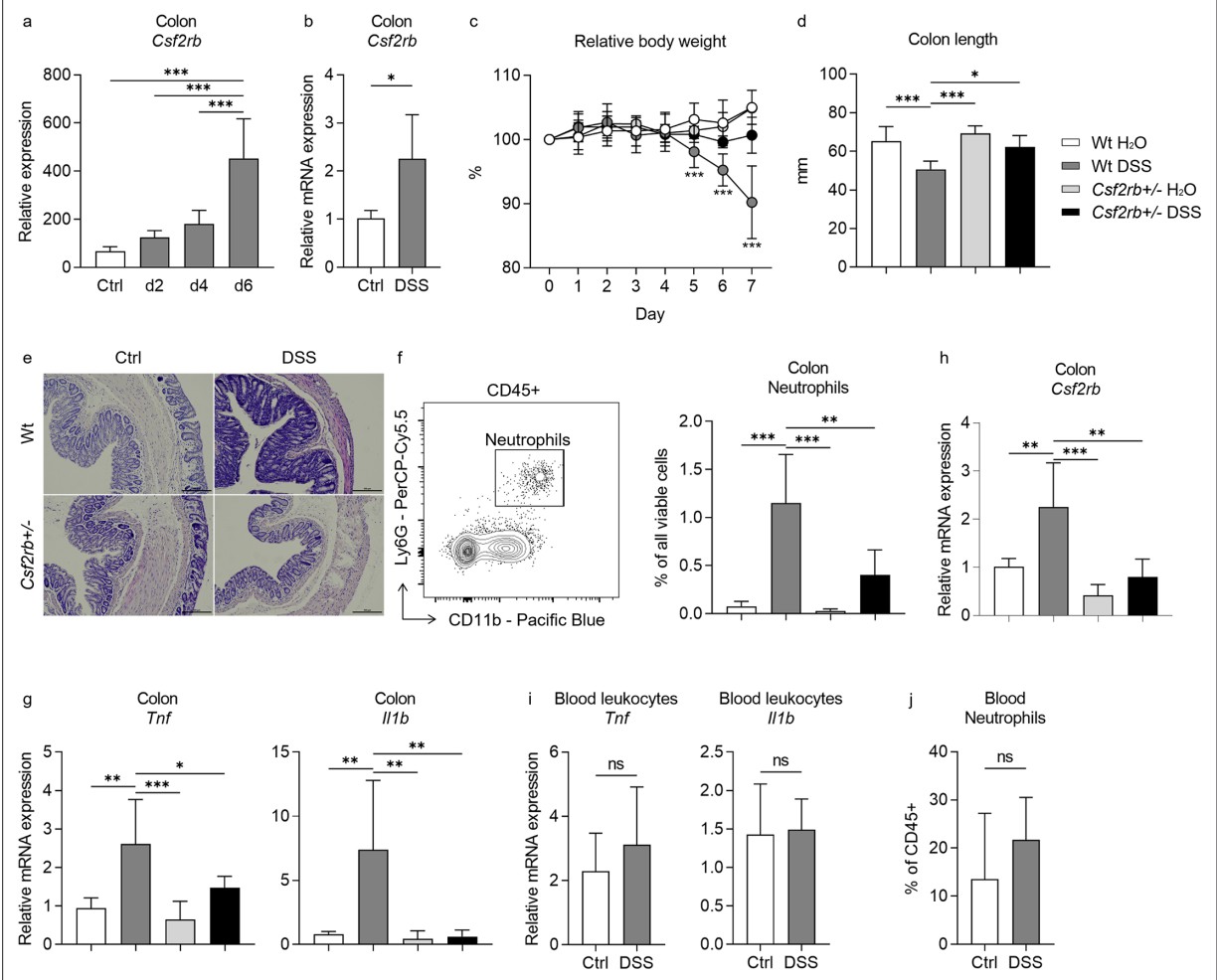

**Figure 1.** CD131 contributed to dextran sulfate sodium (DSS)-induced murine colitis. (**a**) Gene expression dataset shows the change of relative *Csf2rb* gene expression in murine colon tissues with time after DSS administration. (**b**) Relative *Csf2rb* mRNA expression level in colon tissues of control and DSS-treated wt mice. Comparison between wt and CD131-deficient mice during steady state (administrating normal drinking water) and DSS-induced colitis, including: (**c**) relative body weight change; (**d**) colon length; (**e**) a group of exemplary hematoxylin and eosin (H&E) histology sections of murine colon showing inflammatory infiltration and tissue destruction; scale bars indicate 100 µm; (**f**) an exemplary graph showing the gating strategy for identifying CD11b⁺Ly6G⁺ neutrophils on multi-color flow cytometry and their relative cell number in the colonic tissues normalized to all viable cells; (**g**) relative *Tnf* and *Il1b* mRNA expression levels in colon tissues; and (**h**) relative *Csf2rb* mRNA expression level in colon tissues. (**i**) Relative *Tnf* and *Il1b* mRNA expression levels in blood leukocytes of control and DSS-treated wt mice. (**j**) Relative CD11b⁺Ly6G⁺ neutrophils in the blood of control and DSS-treated wt mice, normalized to all viable CD45⁺ leukocytes. ***$p < 0.001$; **$p < 0.01$; *$p < 0.05$; ns, not significant.

The online version of this article includes the following figure supplement(s) for figure 1:

**Figure supplement 1.** CD131 contributed to DSS-induced murine colitis.

**Figure supplement 2.** CD131 contributed to DSS-induced murine colitis.

supplement 1f) and a significant increase in *Il3ra* (*Figure 1—figure supplement 1g*), which encode the α subunit of GM-CSF and IL-3 receptors, respectively, in the colon tissues at day 6 of DSS administration. Whereas, we did not find any change in *Csf2ra* mRNA expression level in the colon tissues of our mice following DSS administration (*Figure 1—figure supplement 1h*), while *Il3ra* mRNA could not be effectively detected (*Figure 1—figure supplement 1i*).

In order to find out the role of CD131 in DSS-induced murine colitis, we subjected CD131-deficient mice (*Csf2rb⁺/⁻*) to DSS administration and compared them to control wt mice. While the wt mice showed reduced body weight, shortened colon length, as well as more tissue destruction seen on hematoxylin and eosin (H&E) staining, increased CD11b⁺Ly6G⁺ neutrophil infiltration, and *Tnf* and *Il1b* mRNA expression in their colon tissues in response to DSS; the CD131-deficient mice were generally protected from

the effects of DSS (*Figure 1c–g*, *Figure 1—figure supplement 2a, b*). These demonstrated that CD131 contributed to the effects of DSS on inducing murine colitis. We could only have heterozygous *Csf2rb* knock-out mice, while the homozygous ones were fatal at an early stage. We observed that the *Csf2rb* mRNA expression was elevated only in the colons of wt mice following DSS treatment, rather than their CD131-deficient counterparts (*Figure 1h*). Nevertheless, the *Tnf* and *Il1b* mRNA expression in blood leukocytes, as well as the neutrophils in the blood, were comparable between DSS-treated and control wt mice (*Figure 1i, j*), indicating a local, but not systemic, inflammation in our model.

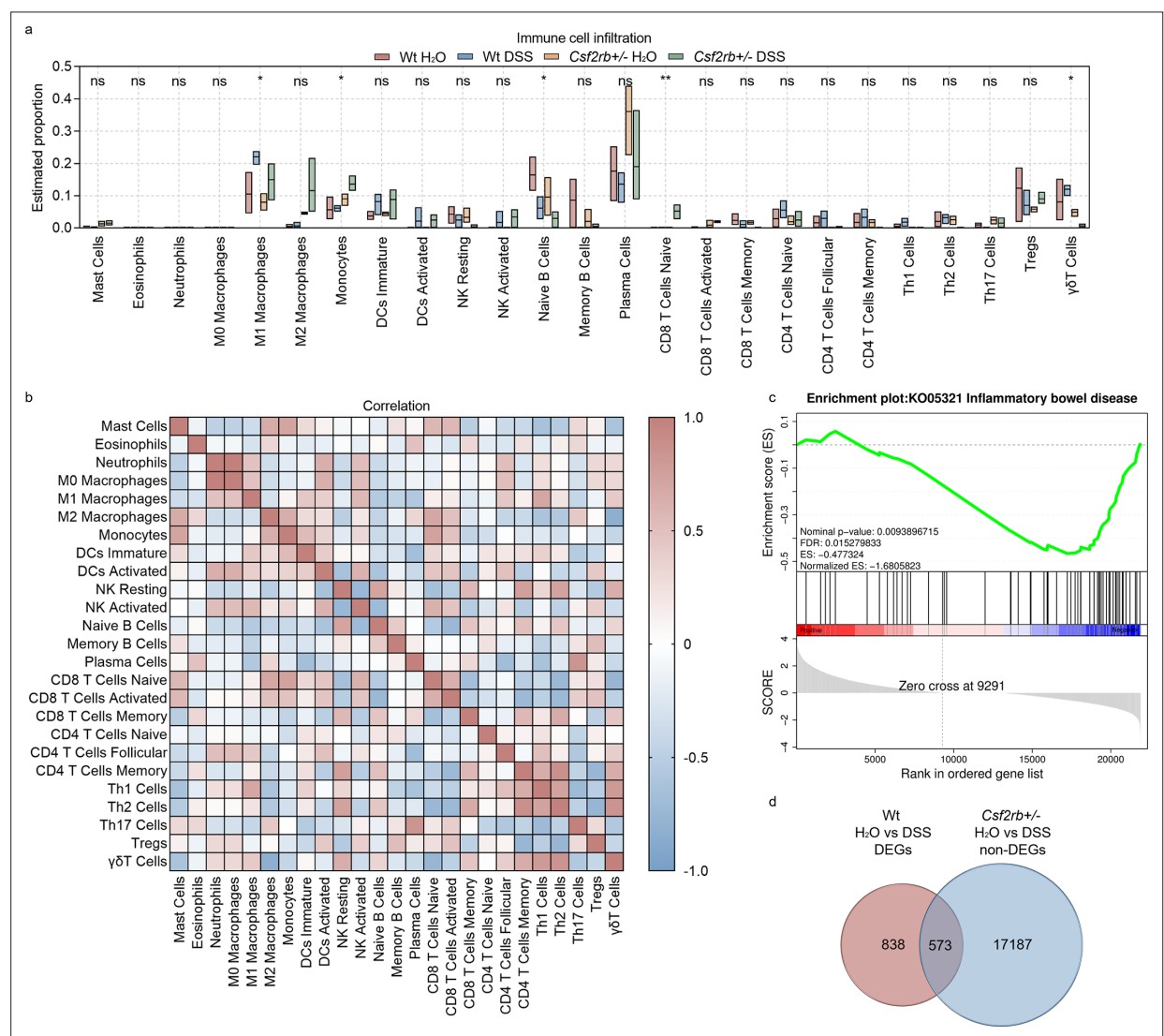

**Figure 2.** CD131 was associated with immune cell infiltration signature. (**a**) Immune cell infiltration analysis of the colon tissues of wt and CD131-deficient mice during steady state and dextran sulfate sodium (DSS)-induced colitis. **p < 0.01; *p < 0.05; ns, not significant. (**b**) Correlation analysis of the estimated infiltrating immune cells. (**c**) Gene set enrichment analysis (GSEA) of colon tissues between wt and CD131-deficient mice showing that inflammatory bowel disease-related pathway was enriched in wt mice. (**d**) Venn graph showing 1411 differentially expressed genes (DEGs) between control and DSS-treated wt mice, as well as 17,760 non-DEGs between control and DSS-treated CD131-deficient mice; the intersection resulted in 573 genes.

The online version of this article includes the following figure supplement(s) for figure 2:

**Figure supplement 1.** CD131 was associated with immune cell infiltration signature.

**Figure supplement 2.** CD131 was associated with immune cell infiltration signature.

## CD131 was associated with immune cell infiltration signature

To explore the possible mechanisms that CD131 involved in, we subjected the colonic tissues from control and DSS-treated, wt and CD131-deficient, mice to RNA-seq. Immune cell infiltration analysis showed that the M1 subtype of macrophages and γδT cells were significantly enriched in the colonic tissues of DSS-treated wt mice, suggesting the association between CD131 and the infiltration of these cell subpopulations (*Figure 2a*). Whereas, the colon of CD131-deficient mice which received DSS exhibited increased monocytes and naive CD8$^+$ T cells, as well as decreased naive B cells. Correlation analyses demonstrated that M1 macrophages were significantly correlated with Th1 cells (rs = 0.6906, p = 0.016); while γδT cells were positively correlated with resting NK cells (rs = 0.6409, p = 0.029), CD4$^+$ memory T cells (rs = 0.6457, p = 0.027), Th1 cells (rs = 0.7988, p = 0.002), and Th2 cells (rs = 0.8451, p = 0.001), but negatively correlated with mast cells (rs = −0.6224, p = 0.035), M2 macrophages (rs = −0.8364, p = 0.001), and naive CD8$^+$ T cells (rs = −0.7619, p = 0.002) (*Figure 2b*).

To investigate the specific pathways that could be associated with CD131, we performed gene set enrichment analysis (GSEA) on the RNA-seq data. We found that IBD-related pathway was enriched in wt mice (*Figure 2c*, *Supplementary file 1*). Besides, several innate and acquired immune-related pathways, including antigen processing and presentation, FcγR-mediated phagocytosis, FcεRI signaling pathway, Toll-like receptor signaling pathway, natural killer cell-mediated cytotoxicity, cell adhesion molecules, TNF signaling pathway, bacterial invasion of epithelial cells, T cell receptor signaling pathway, Th1 and Th2 cell differentiation, Th17 cell differentiation, and B cell receptor signaling pathway, were also enriched in wt mice (*Figure 2—figure supplement 1*, *Supplementary file 1*).

To explore the biological functions of the genes influenced by CD131, we investigated the differentially expressed genes (DEGs) between wt groups and intersected them with the non-DEGs between CD131-deficient groups. There were 1411 DEGs between control and DSS-treated wt mice, as well as 17,760 non-DEGs between the CD131-deficient mice; the intersection resulted in 573 genes (*Figure 2d*, *Supplementary file 2*), comprising 327 up-regulated and 246 down-regulated genes, for further analyses. The Gene Ontology enrichment analysis showed that the up-regulated genes were associated with functions of plasma membrane and synapses (*Figure 2—figure supplement 2a*, *Supplementary file 3*), while the down-regulated genes were associated with cellular and nuclear activities (*Figure 2—figure supplement 2b*, *Supplementary file 4*). Moreover, the Kyoto Encyclopedia of Genes and Genomes (KEGG) enrichment analysis showed that the up-regulated genes were involved in the protein digestion and absorption, cholinergic synapse, and PI3K–Akt signaling pathways (*Figure 2—figure supplement 2c*, *Supplementary file 5*); whereas, the down-regulated genes were involved in the cell cycle, progesterone-mediated oocyte maturation, Fanconi anemia, and DNA replication pathways (*Figure 2—figure supplement 2d*, *Supplementary file 6*).

## Macrophages and CD131 synergistically contributed to DSS-induced murine colitis

As RNA-seq analyses suggested that macrophages were influenced by CD131 expression, we investigated their role in DSS-induced murine colitis. The CD11b$^+$MHCII$^+$F4/80$^+$ macrophages were elevated in the colonic tissues of DSS-treated wt mice, rather than their CD131-deficient counterparts (*Figure 3a*). Besides, CD131 was expressed on these macrophages (*Figure 3b*). Then, we subjected wt and CD131-deficient mice to clodronate liposomes injection to deplete their monocytes/macrophages, and observed their response to DSS administration. The wt mice receiving clodronate liposomes exhibited less body weight loss and colon shortening, as well as reduced macrophage infiltration and *Tnf* mRNA expression in their colon tissues, comparing to those receiving Phosphate-buffered saline (PBS) liposomes (control); whereas, the CD131-deficient mice did not show any response to either liposomes or DSS treatment (*Figure 3c–f*). Besides, the colon-infiltrating macrophages from wt mice expressed elevated level of TNF in response to DSS administration; while those from CD131-deficient mice did not show such alteration (*Figure 3g*). To validate the effect of CD131 signaling, we subjected cultured RAW 264.7 murine macrophage cell line to rm-GM-CSF treatment. Cells treated with rm-GM-CSF exhibited higher *Tnf* mRNA expression and lower 5-(and 6)-Carboxyfluorescein

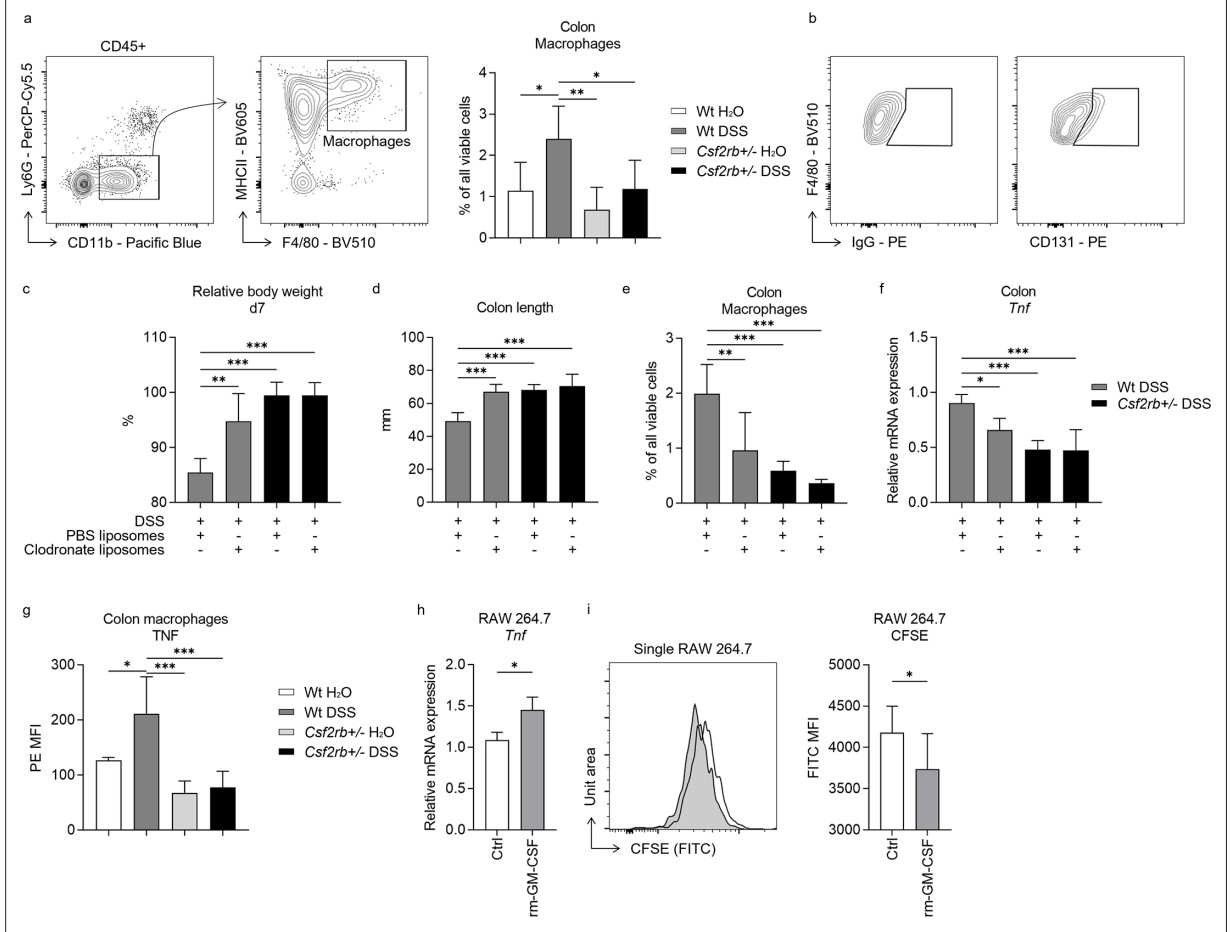

**Figure 3.** Macrophages and CD131 synergistically contributed to dextran sulfate sodium (DSS)-induced murine colitis. (**a**) Exemplary graphs showing the gating strategy for identifying CD11b⁺MHCII⁺F4/80⁺ macrophages on multi-color flow cytometry and their relative cell number, normalized to all viable cells, in the colonic tissues of wt and CD131-deficient mice treated with normal drinking water or DSS. (**b**) Exemplary graphs showing the gating strategy for identifying CD131⁺F4/80⁺ macrophages. Comparison between wt and CD131-deficient mice treated with DSS plus either PBS liposomes or clodronate liposomes, including: (**c**) relative body weight at day 7; (**d**) colon length; (**e**) relative number of macrophages in the colonic tissues normalized to all viable cells; and (**f**) relative *Tnf* mRNA expression level in colon tissues. (**g**) Multi-color flow cytometry showing TNF mean fluorescence intensity (MFI) on colon macrophages in wt and CD131-deficient mice treated with normal drinking water or DSS. RAW 264.7 murine macrophage cell line treated with control or rm-GM-CSF, and relative *Tnf* mRNA expression level (**h**) and CFSE MFI (**i**) were observed. ***p < 0.001; **p < 0.01; *p < 0.05.

The online version of this article includes the following figure supplement(s) for figure 3:

**Figure supplement 1.** Macrophages and CD131 synergistically contributed to DSS-induced murine colitis.

**Figure supplement 2.** Macrophages and CD131 synergistically contributed to DSS-induced murine colitis.

diacetate succinimidyl ester (CFSE) fluorescence intensity as compared with control (**Figure 3h, i**), indicating enhanced pro-inflammatory status and increased proliferation, respectively. These suggested that macrophages mediated the effect of CD131 on contributing to DSS-induced murine colitis, and CD131 might also have mediated the pro-inflammatory effect of macrophages.

We also investigated several other immune cell subpopulations in the colonic tissues (**Figure 3— figure supplement 1**), but their association with CD131 expression during DSS-induced colitis was not as clear as macrophages. Of note, however, we found that CD131 was expressed at different levels on various cell types (**Figure 3—figure supplement 2**), including neutrophils, CD45⁺CD11b⁺MHCII⁺CD11c⁺ cells, CD45⁺CD11b⁻Ly6C⁺ cells, CD3⁺ T cells, CD19⁺ B cells, EpCAM⁺ epithelial cells and EpCAM⁻CD45⁻ cells.

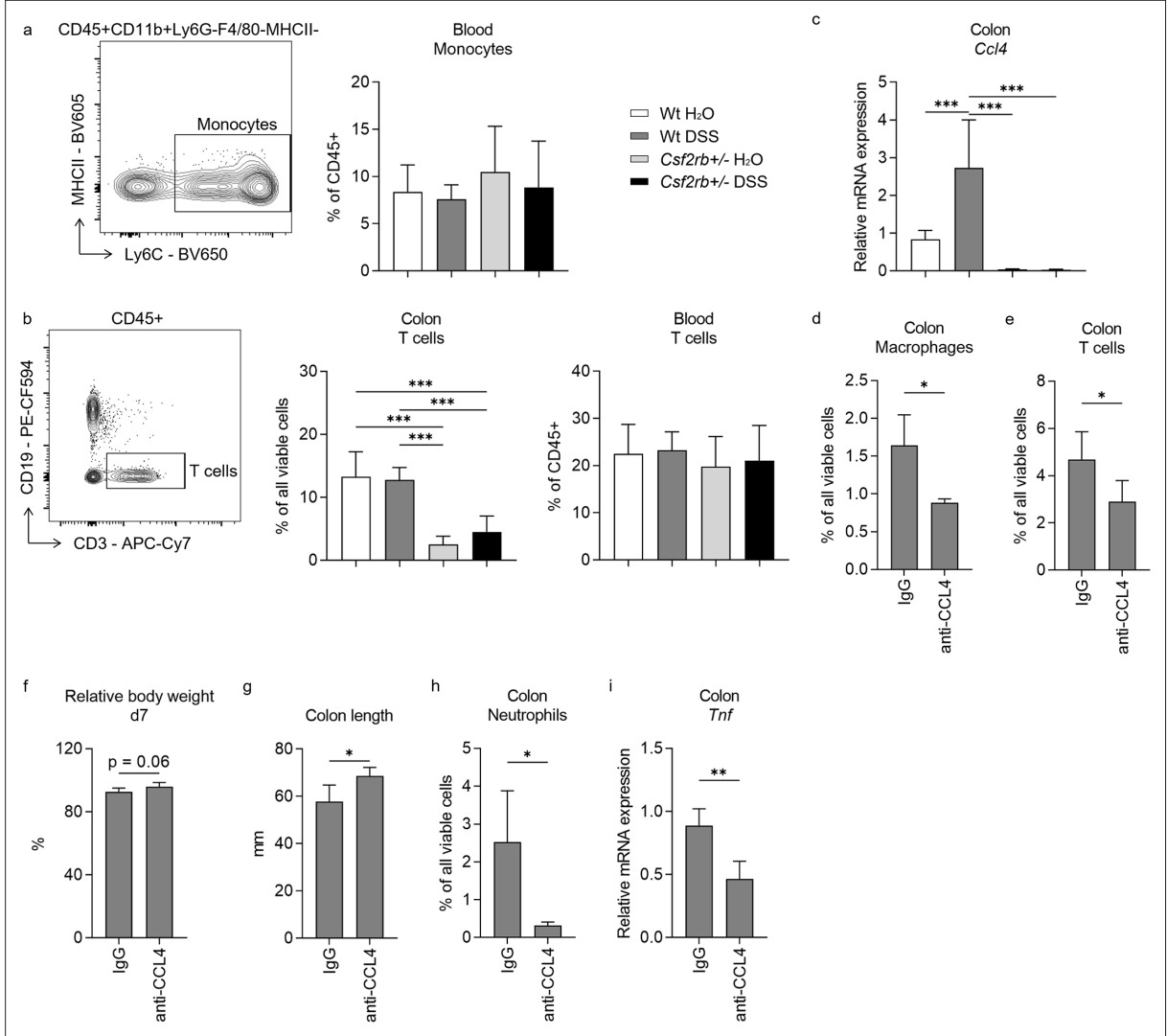

**Figure 4.** CCL4 mediated the effect of CD131 on chemotaxis and inflammatory response. Comparison between wt and CD131-deficient mice treated with normal drinking water or dextran sulfate sodium (DSS), including: (**a**) An exemplary graph showing the gating strategy for identifying CD11b+Ly6C+ monocytes on multi-color flow cytometry, and their relative cell number in the blood normalized to all viable CD45+ leukocytes; (**b**) an exemplary graph showing the gating strategy for identifying CD3+ T cells on multi-color flow cytometry, as well as their relative cell number in the colonic tissues normalized to all viable cells, and in the blood normalized to all viable CD45+ leukocytes; (**c**) relative *Ccl4* mRNA expression in the colon tissues. Comparison between wt mice treated with DSS plus IgG isotype control and anti-CCL4 antibodies, including: (**d**) relative macrophage number in the colon tissues normalized to all viable cells; (**e**) relative T cell number in the colon tissues normalized to all viable cells; (**f**) relative body weight at day 7; (**g**) colon length; (**h**) relative neutrophil number in the colon tissues normalized to all viable cells; and (**i**) relative *Tnf* mRNA expression in the colon tissues. ***p < 0.001; **p < 0.01; *p < 0.05.

The online version of this article includes the following figure supplement(s) for figure 4:

**Figure supplement 1.** Relative Ccl5 mRNA expression levels in colon tissues of wt and CD131-deficient mice treated with normal drinking water or dextran sulfate sodium (DSS).

## CCL4 mediated the effect of CD131 on chemotaxis and inflammatory response

Unlike the change of macrophages in the colonic tissues of wt mice in response to DSS treatment, we observed no difference in CD11b+Ly6C+ monocytes in their blood among different groups of mice (*Figure 4a*). Besides, we found higher level of CD3+ T cell infiltration in the colonic tissues of wt mice, either control or DSS-treated, comparing to CD131-deficient mice; however, the T cells were similar in the blood among the study groups (*Figure 4b*). These suggested that CD131 might have an effect

on the chemotaxis of immune cells to the colon. The RNA-seq data suggested that CCL4 and CCL5 were the only chemotactic factors influenced by CD131 (*Supplementary file 2*). While we found that *Ccl4* mRNA was elevated in the colonic tissues of wt mice treated with DSS comparing to control and CD131-deficient mice (*Figure 4c*), which was consistent with the change observed in RNA-seq data, *Ccl5* did not show such alteration (*Figure 4—figure supplement 1*). To explore if CCL4 was implicated in the chemotaxis of macrophages and T cells into the inflammatory colon, we subjected the wt mice to anti-CCL4 antibody injection to antagonize its effect. Mice treated with anti-CCL4 antibody exhibited reduced macrophage and T cell infiltration into the colonic tissues in response to DSS administration as compared with those treated with IgG isotype control (*Figure 4d, e*). Besides, we found that these mice showed a tendency of less body weight loss (p = 0.06; *Figure 4f*), less colon shortening (*Figure 4g*), as well as lower neutrophil infiltration (*Figure 4h*) and *Tnf* mRNA expression (*Figure 4i*) in the colonic tissues in response to DSS administration, indicating a suppressed inflammatory response in these mice. These suggested that CCL4 might have mediated the effect of CD131 on chemotaxis of macrophages and T cells into the colon, as well as on inflammatory response in DSS-induced murine colitis.

## CD131 was associated with endoscopic and pathological severity of UC as well as macrophage and T cell infiltration into the colon

To validate the role of CD131 in intestinal inflammation as demonstrated by mouse models, we firstly explored the change of CSF2RB gene expression in colonic tissues from a human IBD cohort by analyzing the gene expression dataset GSE179285 (*Keir et al., 2021*). The expression of CSF2RB was elevated in the inflamed colonic tissues from UC patients as compared to the uninflamed colonic tissues from UC patients or healthy controls (*Figure 5a*). Next, we retrospectively analyzed clinical data and colonic tissue specimens from a cohort of UC patients. The demographic characteristics of the patients are summarized in *Table 1*, while the demographic and clinical data are available in *Supplementary file 7*. The tissue damage and inflammatory cell infiltration were evaluated by H&E staining; while CD131 expression, together with macrophage and T cell infiltration, in the colonic tissues were determined by immunohistochemistry (IHC) staining (*Figure 5b–e*). We did not find any correlation between CD131 expression in the colonic tissues and clinical parameters of the patients (*Table 1*). Nevertheless, CD131 expression was significantly correlated with the mucosal appearance evaluated under endoscopy ($\tau$ = 0.378, p = 0.023). Besides, CD131 expression was significantly correlated with Robarts histopathology index (rs = 0.4983, p = 0.006; *Figure 5f*), which reflects the histopathological severity of the intestinal inflammation. In addition, we found that CD131 expression was also correlated with CD68 (rs = 0.7734, p < 0.001; *Figure 5g*) and CD3 (rs = 0.3616, p = 0.05; *Figure 5h*), indicating the infiltration of macrophages and T cells, respectively. These suggested that CD131 was associated with the endoscopic and histopathological severity of the intestinal inflammation, as well as macrophage and T cell infiltration in the colon during UC, reflecting its clinical significance.

## Discussion

CD131 is a receptor subunit mediating the effects of hematopoietic growth factors GM-CSF and IL-3, which regulate various inflammatory responses (*Dougan et al., 2019*). The pleiotropic effects of the cytokines on intestinal inflammation suggest that additional factors influence their overall function, where the receptor may play a role. In the present study, we investigated the role of CD131 in the pathogenesis of UC, with the use of murine colitis model established by administration of DSS in the drinking water. By comparing the immune and inflammatory responses between wt and CD131-deficient mice, we found that CD131 contributed to DSS-induced murine colitis, which functioned in synergy with tissue-infiltrating macrophages. Besides, we also demonstrated that the effect of CD131 on chemotaxis of macrophages, together with T cells, was mediated by CCL4. In addition, we analyzed clinical data and pathology specimens from UC patients and found that CD131 was associated with the endoscopic and pathological severity of intestinal inflammation, as well as macrophage and T cell infiltration. Taken together, we demonstrated the pro-inflammatory effect of CD131 on murine colitis, which may have clinical significance.

There is currently no published study to date dedicated to investigating the role of CD131 in IBD. In a recent study investigating its ligand cytokine, IL-3, in the pathogenesis of UC, the authors showed

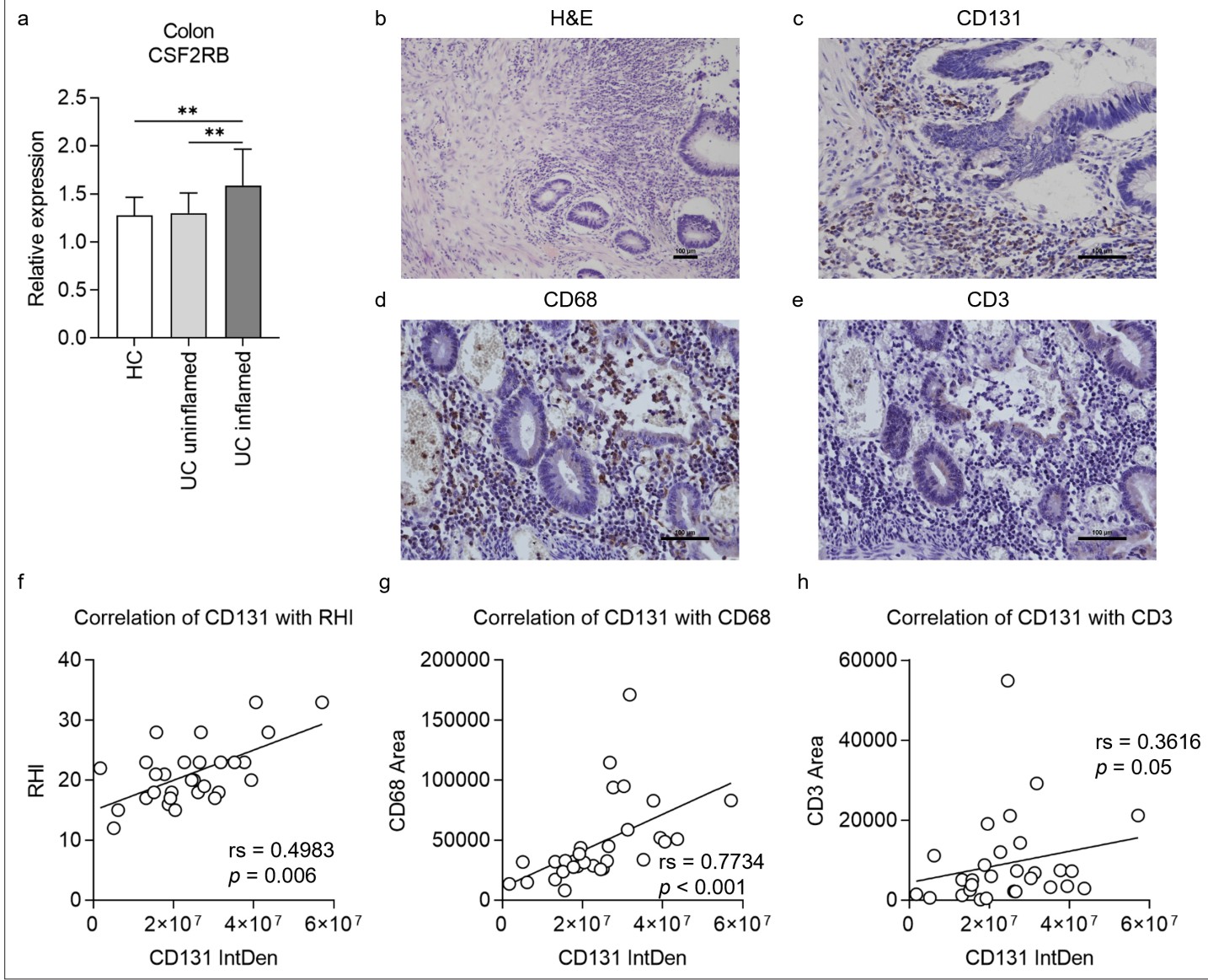

**Figure 5.** CD131 was associated with endoscopic and pathological severity of ulcerative colitis (UC) as well as macrophage and T cell infiltration into the colon. (**a**) Gene expression dataset shows relative CSF2RB gene expression in colonic tissues of healthy controls (HC), uninflamed UC patients and inflamed UC patients. **p < 0.01. A group of exemplary graphs showing histology sections of colonic tissues from UC patients stained with hematoxylin and eosin (H&E) (**b**), CD131 (**c**), CD68 (**d**) and CD3 (**e**); scale bars indicate 100 µm. (**f**) Immunohistochemistry (IHC) analysis showing correlation between CD131 integrated density (IntDen), reflecting its expression level, and Robarts histopathological index (RHI) on H&E, reflecting disease severity. (**g**) IHC analysis showing correlation between CD131 IntDen and CD68 area, reflecting relative positive macrophage cell number. (**h**) IHC analysis showing correlation between CD131 IntDen and CD3 area, reflecting relative positive T cell number.

that CD131 knock-out mice exhibited depressed neutrophil infiltration into the colon in response to DSS administration,(***Bénard et al., 2023***) which was in accordance with our findings. Reduced neutrophil infiltration indicated milder inflammatory response; and we also showed less body weight loss and shortened colon length, less tissue destruction on histology sections, as well as lower inflammatory cytokine expression in the colonic tissues in CD131-deficient mice, which all suggested that CD131 contributed to the intestinal inflammation. Interestingly, however, another study found that activating the heteromer receptor formed by CD131 and erythropoietin receptor could exert potent anti-inflammatory effects and thus ameliorate experimental colitis in mice (***Nairz et al., 2017***). Obviously, the conversion from pro- to anti-inflammatory effects was attributable to the interaction between CD131 and erythropoietin receptor, and the underlying mechanism warrants further investigation.

**Table 1.** Demographic characteristics of ulcerative colitis patients and correlation with CD131 expression.

| | | UC patients | Correlation with CD131 expression | |
|---|---|---|---|---|
| | | | rs or *τ* | p |
| *n* | | 29 | - | |
| Age – year [median (range)] | | 53 (35–80) | 0.112 | 0.564 |
| Gender – *n* (%) | Female | 13 (44.8) | 0.172 | 0.273 |
| | Male | 16 (55.2) | | |
| Disease duration – months [median (range)] | | 48 (1–312) | −0.071 | 0.716 |
| Smoking history – *n* (%) | Never | 20 (69.0) | 0.164 | 0.284 |
| | Former | 7 (24.1) | | |
| | Current | 2 (6.9) | | |
| Fever* – *n* (%) | No | 29 (100) | - | |
| Pulse – *n* (%) | ≤90 | 24 (82.8) | 0.082 | 0.603 |
| | >90 | 5 (17.2) | | |
| Diarrhea – *n* (%) | No | 5 (19.2) | 0.016 | 0.922 |
| | Yes | 21 (80.8) | | |
| | *Unknown* | *3* | | |
| Blood in stool – *n* (%) | Fobt+ | 5 (17.2) | 0.036 | 0.817 |
| | Visible | 24 (82.8) | | |
| Montreal Classification for disease extent† – *n* (%) | E2 | 3 (10.3) | 0.090 | 0.567 |
| | E3 | 26 (89.7) | | |
| WBC‡ – *n* (%) | Low | 5 (17.2) | 0.138 | 0.358 |
| | Normal | 17 (58.6) | | |
| | Elevated | 7 (24.1) | | |
| Anemia** – *n* (%) | No | 2 (6.9) | 0.095 | 0.547 |
| | Yes | 27 (93.1) | | |
| Elevated ESR§ – *n* (%) | No | 8 (50.0) | 0.297 | 0.172 |
| | Yes | 8 (50.0) | | |
| | *Unknown* | *13* | | |
| Mucosal appearance at endoscopy¶ – *n* (%) | Moderate | 4 (15.4) | 0.378 | **0.023** |
| | Severe | 22 (84.6) | | |
| | *Unknown* | *3* | | |

Italics indicate data not analyzed.

*Fever was defined as body temperature higher than 37.3°C.

†Montreal Classification for disease extent: E1, involvement limited to the rectum (i.e., proximal extent of inflammation is distal to the rectosigmoid junction); E2, involvement limited to the portion of the colorectum distal to the splenic flexure; E3, involvement extends proximal to the splenic flexure.

‡Blood WBC count between $(4.5–11) \times 10^9$ /l was considered normal.

§Erythrocyte sedimentation rate (ESR) was considered elevated if >15 mm/hr in male or >20 mm/hr in female patients.

¶Mucosal appearance at endoscopy was evaluated as per the Mayo score, including normal or inactive disease; mild disease (erythema, decreased vascular pattern, mild friability); moderate disease (marked erythema, absent vascular pattern, friability, erosions), and severe disease (spontaneous bleeding, ulceration). Fobt: fecal occult blood test.

**Anemia was defined as blood hemoglobin less than 13.5 g/dl in male or 12.0 g/dl in female patients.

One concern about our findings was that our CD131-deficient mice were all heterozygous, while homozygous ones were fatal at an early stage. It appears that homozygous CD131-deficient mice were used by others;(*Nairz et al., 2017*; *Rousselle et al., 2022*) the gene knock-out manipulation of our mice might be different from theirs which accounts for the early fatality, but it is unknown. Nevertheless, we observed significantly lower *Csf2rb* mRNA expression in these mice, and performed repeated experiments to confirm our findings.

Macrophages are key players in inflammation which exert both pathogenic and protective functions depending on the subset differentiation and polarization in response to local microenvironment signals (*Murray and Wynn, 2011*). We showed that macrophages were elevated in the colonic tissues of DSS-treated wt mice, while depleting macrophages with clodronate liposomes was protective with respect to body weight loss, colon shortening, and inflammatory cytokine expression. These were consistent with the previous studies showing that macrophages contributed to murine colitis (*Pan et al., 2022*). Of note, however, studies also indicated that M2-like macrophages were also present in the intestine to alleviate inflammation.(*Bain and Mowat, 2014a*) As simply defining the macrophages as M1 or M2 phenotypes is not readily suitable in the intestine (*Bain and Mowat, 2014a*), we demonstrated elevated expression of TNF in intestinal macrophages in DSS-treated wt mice, suggesting a pro-inflammatory role and possibly predominance of M1-like phenotype in murine colitis. Besides, we showed that CD131 was expressed on colonic macrophages, as is already known,(*Becher et al., 2016*; *Dougan et al., 2019*) which allows them to respond to the stimulation by cytokines GM-CSF and IL-3. By stimulating cultured murine macrophage cell line with rm-GM-CSF, we demonstrated elevated TNF expression, which indicated that CD131 signaling could exert pro-inflammatory effect. Thus, our results suggested that macrophages mediated the pro-inflammatory effect of CD131 on DSS-induced colitis in wt mice.

On the other hand, our CD131-deficient mice did not exhibit elevation in colon macrophages upon DSS administration, nor did these macrophages produce more TNF. Besides, the changes following macrophage depletion, as seen in wt mice, were absent in the CD131-deficient counterparts subjected to DSS administration. It is reasonable to believe that, as a result of lacking CD131, these mice were not able to effectively respond to GM-CSF and IL-3 stimulation. Therefore, the macrophage-mediated inflammatory response in the colonic tissues of DSS-treated wt mice was, at least in part, attributable to CD131 signaling. Our findings were consistent with previous studies showing that GM-CSF promoted pro-inflammatory function of macrophages and thus facilitated intestinal inflammation by various mechanisms (*Castro-Dopico et al., 2020*; *Zhang et al., 2020*; *Griseri et al., 2012*), although these studies were not dedicated to the investigation of the function of CD131. Admittedly, our findings were limited by the fact that we did not have macrophage-specific CD131-deficient mice, therefore we could not confirm the effect of macrophage-specific CD131 on DSS-induced murine colitis, which awaits further investigations.

As macrophages and T cells were higher in the colonic tissues, but not in the blood, of DSS-treated wt mice instead of CD131-deficient ones, we speculated that a possible mechanism affecting the chemotactic process was involved. It is consistent with a previous study demonstrating that the circulating monocytes migrate to the intestine to maintain a normal intestinal macrophage pool (*Bain et al., 2014b*). CCL4, previously known as macrophage inflammatory protein-1β (MIP-1β), is a chemokine produced in high amounts by monocytes and macrophages (*Menten et al., 2002*). CCL4 signals through its receptor CCR5, and is involved in the chemotaxis and trans-endothelial migration of monocytes, T cells, and other immune cells, which possibly contributes to local inflammatory response. It is in accordance with our findings that antagonizing CCL4 with antibodies resulted in decreased macrophage and T cell infiltration into the colon and alleviation of intestinal inflammation. We showed that the expression of CCL4 was elevated in colonic tissues of DSS-treated wt mice, but not their CD131-deficient counterparts. To the best of our knowledge, there is so far no study investigating the effect of CD131 signaling on chemokines like CCL4. One of the downstream signaling mechanisms of GM-CSF/IL-3 receptor complexes is the JAK2/STAT5 pathway, which controls the differentiation and inflammatory signature of immune cells (*Becher et al., 2016*; *Dougan et al., 2019*). A previous study demonstrated that blocking SOCS3 on macrophages, a feedback inhibitor protein of JAK/STAT signaling pathway, resulted in enhanced and prolonged activation of JAK/STAT pathway, as well as elevated expression of M1 phenotype-associated genes (*Qin et al., 2012*). Notably, the authors showed that mice with SOCS3-deficient macrophages experienced exacerbated Lipopolysaccharide (LPS)-induced

sepsis, which was associated with enhanced JAK/STAT pathway activation and increased plasma levels of cytokines/chemokines such as CCL4. Therefore, their results appear to support our findings that CD131 signaling could possibly contribute to the elevation of CCL4 expression. Of note, however, CD131 signaling might also contribute to the local proliferation of macrophages, as indicated by the accelerated proliferation of cultured macrophage cell line in response to GM-CSF stimulation, but apparently it warrants future investigation in colitis.

The major limitation of the present study is that we did not investigate the role of CD131 expressed on other immune cell populations. Apart from monocytes and macrophages, we also showed that CD131 was expressed at different levels on various cell types, as is already known from previous studies (*Becher et al., 2016*; *Dougan et al., 2019*). Although the role of these cells in CD131-contributed intestinal inflammation needs to be investigated in future studies, the effect of CD131, in synergy with macrophages, on murine colitis is still important as shown in the present study. Besides, the functional status of CD131 molecule was not studied in the present work. Its ligand cytokine GM-CSF appears to exert pleiotropic effects in the intestine: although GM-CSF is generally accepted as a pro-inflammatory cytokine contributing to colitis, treatment with a recombinant human GM-CSF agent, sargramostim, has shown some benefit in Crohn's disease patients.(*Dieckgraefe and Korzenik, 2002*; *Korzenik et al., 2005*; *Valentine et al., 2009*) This inconsistency raised the possibility that the functional status of the receptor molecules might play a role in the overall effect of GM-CSF stimulation. In support of this, a genetic analysis of Ashkenazi Jewish populations, which show a high prevalence of Crohn's disease, demonstrates that monocytes from carriers of a frameshift mutation in CSF2RB gene exhibit reduced responses to GM-CSF treatment (*Chuang et al., 2016*). It suggests that an intact wild-type CD131 molecule is essential in mediating proper response to GM-CSF treatment; nevertheless, the underlying mechanisms await further studies. Lastly, the present study may be limited by the low number of UC patients included in the analysis. Although we have a high volume of UC patients in our medical center, the need for surgical intervention is significantly decreased since the advent of biologicals. Those who did receive surgical treatment were generally suffering from severe colitis, which could explain the reason why CD131 expression had no significant correlation with clinical evaluation. Therefore, caution need to be exercised when interpreting the results of our patient cohort analysis.

In conclusion, the present study demonstrated that CD131 and tissue-infiltrating macrophages synergistically contributed to DSS-induced murine colitis. Besides, CD131 may have promoted the chemotaxis of macrophages and T cells into the colon through CCL4. In addition, CD131 was associated with the endoscopic and pathological severity of intestinal inflammation in clinical UC patients. The present study provides a novel way to the understanding of the mechanisms of GM-CSF and IL-3 effects in the intestine, which will benefit the development of therapeutic approaches. For instance, as discussed above, IBD patients with a functionally intact wild-type CD131 receptor molecule may have better clinical responses to sargramostim treatment and should be selected for future clinical trials. Indeed, more insightful studies are needed to improve the understanding of GM-CSF-CD131 signaling in the pathogenesis of intestinal inflammation, which may shed light on an alternative avenue of IBD therapy avoiding immunosuppression.

## Materials and methods
### Human
This observational study was performed in accordance with the Declaration of Helsinki. All protocols involving human subjects were approved by the Institutional Review Board (#2023PS953K), while written informed consent was exempted as only retrospective, de-identified clinical data and archival pathology specimens were used without jeopardizing the clinical treatment of the patients. Patients aged 18 years or greater, who underwent elective surgery and had a pathological diagnosis of UC, at Shengjing Hospital of China Medical University between 2013 and 2023, were retrospectively included. The clinical data, including age, gender, disease duration, smoking history, body temperature, pulse, diarrhea, blood in stool, disease extent, white blood cell count, hemoglobin, erythrocyte sedimentation rate, and mucosal appearance at endoscopy were retrieved from the clinical registry. The disease extent was evaluated according to the Montreal Classification for IBD (*Satsangi et al., 2006*), while the mucosal appearance at endoscopy was evaluated according to the endoscopic section of the Mayo score (*D'Haens et al., 2007*). In addition, colonic tissue sections from pathology

specimens were obtained from the Department of Pathology, stained with H&E and IHC, and analyzed in correlation with the clinical data.

## Mice

C57BL/6J mice (wt) were obtained from Beijing HFK Bioscience Co, Ltd, China. CD131-deficient mice (*Csf2rb*$^{+/-}$) on a C57BL/6J background were purchased from Shanghai Model Organisms Center, Inc, China (C57BL/6JSmoc-*Csf2rb*$^{em1Smoc}$, #NM-KO-190787). All mice were bred in specific-pathogen-free (SPF) animal facility, and were 8–12 weeks of age at the time of euthanasia. All animal studies were performed in an age- and sex-matched manner and littermate controls were adopted. At least 5 mice per group were used for all experiments. All protocols were approved by the Institutional Animal Care and Use Committee (#2022PS856K).

## Chemically induced experimental colitis and in vivo interventions

### DSS-induced murine colitis

Acute DSS-induced colitis was established as per previously described protocols (*Wirtz et al., 2017*). Briefly, 8- to 10-week-old SPF mice at least 18 g of weight were given 2% (wt/vol) DSS (M.Wt. 36,000–50,000 Da; #160110, MP Biomedicals, USA) in the drinking water for 7 days. The DSS solutions were freshly prepared daily, and the volume of leftover DSS solution was measured the second day to exclude the possibility that changes in colitis activity were due to differences in DSS consumption. No difference in water consumption was observed between CD131-deficient and wt mice. The body weight was recorded on a daily basis until termination of the experiment.

### Monocyte depletion

To deplete monocytes, 200 µl clodronate liposomes (Liposoma BV, The Netherlands) were injected into the peritoneum 3 days before and 2 days after the start of DSS administration, and control mice were injected with 200 µl PBS liposomes (Liposoma BV).

### Anti-CCL4

To antagonize the effect of CCL4, 200 µg anti-mouse CCL4 antibody (BioLegend, USA) in 200 µl PBS was injected into the tail vein of wt mice on the day of and 3 days after the start of DSS administration, and control mice were injected with 200 µg IgG2b-κ isotype control antibody (BioLegend).

## Tissue harvesting and cell isolation

Upon termination of the experiment, the mice were euthanized and the entire colon and blood were sampled.

### Colon

The entire colon was harvested and its length and weight were determined. After flushing out the stools with PBS, colonic segments 0.5 cm in length from both the proximal and distal parts were sampled. The colonic tissues for cell isolation were mechanically minced into small pieces and subjected to enzymatic digestion with 450 U/ml collagenase I (Sigma-Aldrich, USA), 60 U/ml collagenase XI (Sigma-Aldrich), 60 U/ml DNase I (GlpBio, USA), and 60 U/ml hyaluronidase (Solarbio, China), at 37°C for 1 hr while shaking at 750 rpm to disrupt the extracellular matrix. Subsequently, the samples were collected and filtered through a 40-µm cell strainer (Biosharp, China) to acquire cell suspension.

### Blood

The blood was collected by aortic puncture, using heparin as the anticoagulant. After removing the plasma, erythrocytes were lysed using RBC lysis buffer (Solarbio) according to the manufacturer's instructions, while leukocytes were collected.

### Intra-peritoneal lavage

The peritoneal cavity was lavaged with 3 ml PBS twice to retrieve leukocytes.

### Mesenteric lymph nodes

The mesenteric lymph nodes were mechanically homogenized and filtered through a 40-μm cell strainer to acquire cell suspension.

### Spleen

The spleen was mechanically homogenized and filtered through a 40-μm cell strainer, while erythrocytes were lysed with RBC lysis buffer (Solarbio) to acquire leukocyte suspension.

### Thymus

The thymus was mechanically disrupted and homogenized. The cell suspensions were centrifuged at $350 \times g$, 4°C for 5 min unless otherwise indicated.

## Cell culture and in vitro interventions

### Cell culture

RAW 264.7 murine macrophages (#SCSP-5036, Shanghai Institute of Biochemistry and Cell Biology, China) were cultured in Dulbecco's Modified Eagle Medium (DMEM) medium (#11995, Solarbio) supplemented with 10% fetal bovine serum (FBS; Procell, China), and kept in a humidified 5% $CO_2$ incubator at 37°C. The FBS used for macrophage culture was heat-inactivated by immersing in 56°C water bath for 30 min. The cells were authenticated by Short tandem repeat (STR) profiling and the mycoplasma contamination was tested negative. All cell assays were performed three times.

### In vitro cytokine stimulation

Cells were seeded at a density of 20,000 cells/100 μl medium in 24-well flat-bottom plates and cultured for 24 hr. For cytokine stimulation, rm-GM-CSF (Peprotech, USA) was used at a final concentration of 20 ng/ml and incubated for 24 hr. PBS containing 5% trehalose was added to the medium of control cells since it was used as the diluting reagent for the recombinant protein.

### Proliferation assay

Cells were labeled with CFSE Cell Division Tracker Kit (BioLegend) according to the manufacturer's instructions, prior to inoculation into the plates. After rm-GM-CSF stimulation for 24 hr, the cells were harvested and fixed with 4% formaldehyde (Solarbio) for 20 min, and the CFSE fluorescence was detected under the FITC channel on an LSRFortessa (BD Biosciences, USA). Data were analyzed with FlowJo v10.8.1 (BD Biosciences).

## Flow cytometry

The following anti-mouse antibodies were used for multi-parameter flow cytometric analyses: anti-CD326 (EpCAM)-APC, G8.8 (eBioscience, USA); anti-CD45.2-FITC, 104 (BioLegend); anti-CD11b-Pacific Blue, M1/70 (BioLegend); anti-Ly-6G-PerCP-Cy5.5, 1A8 (BD Biosciences); anti-F4/80-Brilliant Violet 510, BM8 (BioLegend); anti-I-A/I-E-Brilliant Violet 605, M5/114.15.2 (BioLegend); anti-Ly-6C-Brilliant Violet 650, HK1.4 (BioLegend); anti-CD3-APC-Cy7, 17A2 (BD Biosciences); anti-CD19-PE-CF594, 1D3 (BD Biosciences); anti-CD11c-PE-Cy7, HL3 (BD Biosciences); anti-CD131-PE, JORO50 (BD Biosciences); anti-TNF-PE, MP6-XT22 (BD Biosciences); anti-IgG1, κ-PE, R3-34 (BD Biosciences); and anti-CD16/CD32-unconjugated, 93 (eBioscience).

### Staining strategies

The cell concentration was adjusted to no more than 1 million cells per 100 μl by re-suspending with appropriate volume of PBS. The cells were firstly stained with Zombie UV Fixable Viability Kit (BioLegend) according to the manufacturer's instructions to exclude non-viable cells. Subsequently, the cells were sequentially incubated with purified anti-CD16/CD32 mAb (1:100) at 4°C for 5 min to block Fc receptors, and with antibody cocktail at 4°C in the dark for 30 min to stain membranous proteins. The antibodies for membranous proteins were diluted at 1:200 in PBS containing 1% FBS (Procell), 0.5% bovine serum albumin (BSA; Solarbio), and 2 mM Ethylenediaminetetraacetic acid (EDTA; Solarbio). Due to an inevitable delay for the analysis at the flow cytometry facility, the cells were fixed after staining with 4% formaldehyde (Solarbio) for 20 min and stored at 4°C protected

from light until analysis. Where intracellular protein detection was indicated, the cells were fixed with 4% formaldehyde (Solarbio) and then permeabilized with Permeabilization Buffer (eBioscience), each for 20 min, following membranous protein staining as described above. Afterwards, the cells were incubated with antibodies (diluted at 1:100 in Permeabilization Buffer) at 4°C in the dark for 30 min to stain intracellular proteins.

## Data analysis

Data were acquired on an LSRFortessa (BD Biosciences) and analyzed with FlowJo v10.8.1 (BD Biosciences). The percentages of cells of interest within a parental population were calculated as appropriate. In addition, mean fluorescence intensity of TNF, indicating its average expression level on a cell population, was calculated for the cell populations of interest. Cells were defined as: neutrophils ($CD45^+CD11b^+Ly6G^+$), macrophages ($CD45^+CD11b^+F4/80^+MHCII^+Ly6G^-$), monocytes ($CD45^+CD11b^+Ly6C^+Ly6G^-F4/80^-MHCII^-$), and T cells ($CD45^+CD3^+CD19^-MHCII^-$).

## Histology

### Mouse

Murine colonic tissues were subjected to formalin-fixed, paraffin-embedded tissue sections as per standard protocol. Sections 4 µm in thickness were prepared and mounted onto the slides. The colon tissue sections were subjected to H&E staining according to standard protocol for evaluating tissue damage and lamina propria inflammatory cell infiltration (*Wirtz et al., 2017*). The sections were imaged at ×100x magnification using an ECLIPSE Ci digital scanner (Nikon, Japan).

### Human

Serial colonic tissue sections were prepared from archival formalin-fixed, paraffin-embedded specimens by an expert pathologist at the Department of Pathology. One section from each patient was subjected to H&E staining according to standard protocol for evaluating tissue damage and inflammatory cell infiltration; while other sections were subjected to IHC staining for evaluating CD131 expression as well as macrophage and T cell infiltration. For IHC staining, the sections were firstly incubated with 3% hydrogen peroxide ($H_2O_2$) in PBS to inactivate endogenous peroxidase, followed by blocking with 10% normal goat serum, each for 40 min at room temperature. Subsequently, the sections were incubated with anti-CD131 (1:200 in PBS, Proteintech, USA), CD68 (1:200 in PBS, clone KP1, eBioscience), or CD3 (1:100 in PBS, clone UCHT1, eBioscience) primary antibodies overnight at 4°C. Afterwards, the sections were sequentially incubated with a biotinylated secondary antibody and streptavidin–horseradish peroxidase, each for 30 min at room temperature; followed by development with 3,3'-diaminobenzidine (1.5 min for CD131 and CD68, 8.5 min for CD3). The reagents for IHC staining, except primary antibodies, were obtained from MXB Biotechnologies, China. The sections were then counterstained with hematoxylin to identify the nuclei. Finally, all slides were coverslipped with an aqueous-based mounting medium, and images were captured at ×100 (for H&E) or ×200 (for IHC) magnification using NIS-Elements F software (v6.4, Nikon) on an ECLIPSE Ci digital scanner (Nikon).

### Human histology sections analysis

The H&E sections were blindly analyzed by L.L. and Q.L. according to the Robarts histopathological index (*Mosli et al., 2017*). The IHC sections were assessed with three field-of-views at ×400 magnification using the IHC Image Analysis Toolbox on the ImageJ software (*Schneider et al., 2012*). The CD131 integrated density (IntDen) for each sample was calculated for evaluating its expression level, while CD68 and CD3 'areas' were calculated which could reflect the relative positive cell number.

## Molecular biology

### Polymerase chain reaction for mice genotyping

Genomic DNA was isolated from murine tail biopsies by enzymatic digestion with Willy buffer [100 mM Tris(hydroxymethyl)aminomethane (Solarbio), 200 mM sodium chloride (Rhawn, China), 10 mM EDTA disodium salt dihydrate (Solarbio), and 0.2% sodium dodecylsulfate (Solarbio)] containing 0.6 mg/ml proteinase K (GlpBio) at 56°C for at least 150 min while shaking at 750 rpm to disrupt the extracellular

matrix. The amplification was performed using the 2× Rapid Taq Master Mix (Vazyme, China) according to the manufacturer's instructions. The amplification products were subjected to 1.5% agarose gel electrophoresis to visualize the DNA bands. The following primer pair (Sangon Biotech, China) was used for genotyping: P1 (fwd): TCA GGA AAC AGA GGC AGG AGG AT, P2 (rev): AGT GGC AGG GGC AGT TTG GTT T. The desired polymerase chain reaction (PCR) product size was 646 bp for wt and 335 bp for *Csf2rb* knock-out alleles. An exemplary graph of wt and heterozygous knock-out genotypes is shown in *Figure 1—figure supplement 2c*.

## Real-time quantitative PCR

Total RNA was isolated from homogenized whole colonic tissue or blood leukocytes using the TRIzol reagent (Vazyme) as per standard protocol. RNA quantity and quality were determined spectrophotometrically on a NanoPhotometer N50 (Implen, Germany). The cDNA was reverse transcribed from 1 μg of total RNA for each sample using the HiScript III RT SuperMix with gDNA wiper (Vazyme) according to the manufacturer's instructions. The amplification was performed in duplicates using the ChamQ Universal SYBR qPCR Master Mix (Vazyme) on a 7500 Fast Real-Time PCR System (Applied Biosystems, USA). Relative gene expression was calculated using the ΔΔCt method and normalized to β-actin (*Actb*). At least three independent samples per group were analyzed. The following primers (Sangon Biotech) were used to detect murine genes: *Actb*-fwd: TGT CCA CCT TCC AGC AGA TGT, *Actb*-rev: AGC TCA GTA ACA GTC CGC CTA GA; *Csf2rb*-fwd: ACA GAG AAC CTA GAT CGA GCC, *Csf2rb*-rev: GTG TAC TCT TCG CTC CAC TTG; *Csf2ra*-fwd: CTG CTC TTC TCC ACG CTA CTG, *Csf2ra*-rev: GAG ACT CGC CGG TGT ATC C; *Il3ra*-fwd: CTG GCA TCC CAC TCT TCA GAT, *Il3ra*-rev: GTC AGC CCA GAC AAA GAT GTC; *Tnf*-fwd: CAG GCG GTG CCT ATG TCT C, *Tnf*-rev: GGC CAT TTG GGA ACT TCT CAT C; *Il1b*-fwd: GAA ATG CCA CCT TTT GAC AGT G, *Il1b*-rev: TGG ATG CTC TCA TCA GGA CAG; *Ccl4*-fwd: TTC CTG CTG TTT CTC TTA CAC CT, *Ccl4*-rev: CTG TCT GCC TCT TTT GGT CAG; *Ccl5*-fwd: GCT GCT TTG CCT ACC TCT CC, *Ccl5*-rev: TCG AGT GAC AAA CAC GAC TGC.

## RNA sequencing

### RNA extraction, library construction, and sequencing

Total RNA was isolated from homogenized murine colonic tissues using TRIzol reagent (Invitrogen, USA). RNA quality was assessed using RNase-free agarose gel electrophoresis on a 2100 Bioanalyzer (Agilent, USA). Afterwards, the mRNA was enriched with Oligo(dT) beads. Subsequently, the enriched mRNA was fragmented by using fragmentation buffer and reverse transcribed into double-stranded cDNA by using NEBNext Ultra RNA Library Prep Kit for Illumina (New England Biolabs, USA) as per the manufacturer's instructions. The resulting ds cDNA was sequentially purified, end-repaired, adenine base-added, and ligated to Illumina sequencing adapters. The ligation reaction was purified with AMPure XP Beads; while the adapter-ligated cDNA was subjected to PCR enrichment and again purification. Finally, the resulting cDNA library was sequenced on a NovaSeq 6000 Sequencing System (Illumina, USA). The original RNA-seq data that support the findings of the present study have been deposited into CNGB Sequence Archive (CNSA) of China National GeneBank DataBase (CNGBdb) with accession number CNP0005868.

### Data processing

Raw sequencing reads were processed with fastp (*Chen et al., 2018*) to remove adapters, unknown nucleotides, and low-quality bases. The reads were then mapped to ribosomal RNA (rRNA) database by using bowtie2 (*Langmead and Salzberg, 2012*) to remove possible rRNA reads; while the remaining clean reads were aligned to the murine reference genome assembly GRCm39 by using HISAT2 (*Kim et al., 2015*) with default parameters. The mapped reads of each sample were assembled with StringTie (*Pertea et al., 2015*) and annotated with gene annotation from Ensembl release 110, followed by normalization to transcripts per kilobase of exon model per million mapped reads (TPM) for quantification of the gene abundance by using RSEM software (*Li and Dewey, 2011*).

## Immune cell infiltration analyses

The estimation of immune cell subset abundances within the tissues based on their gene expression profiles was performed by the cell-type identification by estimating relative subsets of RNA transcripts (CIBERSORT) method (*Newman et al., 2015*) on R statistical software v4.3.3 (*R Development Core Team, 2023*). The expression profile of marker genes for the murine immune cell populations was referenced to the results of *Chen et al., 2017*. The Spearman correlation coefficient (rs) was calculated to indicate the relationship between different infiltrating immune cells.

## Gene set enrichment analysis

GSEA (*Subramanian et al., 2005*) was performed to identify the pathways that were significantly different between two study groups by using the clusterProfiler package (v4.12.2) (*Wu et al., 2021*) on R. The input genes were ranked by the Signal2Noise method. The gene sets database 'm2.cp.v2023.2.Mm.symbols.gmt' was chosen as the reference database. A pathway with false discovery rate (FDR) <0.05 was considered as statistically significant.

## DEGs analyses

The identification of DEGs between two study groups was performed by calculating TPM differences using limma package (*Ritchie et al., 2015*). Genes with FDR <0.05 and absolute $\log_2$(fold change [FC]) >1 were considered as DEGs. As we were investigating the role of CD131 on murine colitis, we intersected the DEGs between wt groups and non-DEGs between CD131-deficient groups, and subjected the acquired genes to further analyses. The Gene Ontology (*Ashburner et al., 2000*) and KEGG (*Kanehisa and Goto, 2000*) enrichment analyses were performed by subjecting the intersect gene lists to DAVID bioinformatics (*Huang et al., 2009*) to explore the main biological functions of these genes. The enrichment was considered significant at FDR <0.05.

## Gene microarray data analyses

We selected a human IBD gene expression dataset GSE179285 (*Keir et al., 2021*) and a DSS-induced murine colitis gene expression dataset GSE22307 (*Fang et al., 2011*) from the Gene Expression Omnibus (GEO) database (*Edgar et al., 2002*). In GSE179285 study, patients with IBD and healthy volunteers from multiple centers in North America were enrolled in separate studies, where in one cohort none of the patients were taking any medications for their disease, and in the other cohort none of the patients received treatment with investigational drugs or biologics within the previous 3 months. The normal intestinal tissues from healthy controls ($n$ = 31), uninflamed ($n$ = 32) and inflamed ($n$ = 23) tissues from UC patients, as well as uninflamed ($n$ = 121) and inflamed ($n$ = 47) tissues from Crohn's disease patients were collected during ileocolonoscopy. Total RNA was extracted from the intestinal tissues and gene expression profile was generated based on Agilent Whole Human Genome Microarray 4x44K G4112F (GPL6480 platform) and uploaded by the research group. The Crohn's disease patients were not included in our analysis. In GSE22307 study, C57BL/6J mice were given 3% DSS in the drinking water and the colon tissues were collected at days 0 ($n$ = 5, control group euthanized prior to DSS treatment), 2 ($n$ = 6), 4 ($n$ = 6), and 6 ($n$ = 6). Subsequently, total RNA was extracted from the colon tissues and gene expression profile was generated based on Affymetrix Mouse Genome 430 2.0 Array (GPL1261 platform) and uploaded by the research group. The normalized expression levels of genes of interest were extracted manually and analyzed.

## Statistics

The Shapiro–Wilk test was used to check for the Gaussian distribution of the data. For comparing between two groups, unpaired $t$ test with or without Welch's correction (when Gaussian distribution was assumed) or Mann–Whitney test (when Gaussian distribution was not assumed) was performed. The p values are two-tailed. For the comparison among three or more groups, the statistical methods included one-way ANOVA with Tukey's multiple comparison correction (when Gaussian distribution was assumed) and Kruskal–Wallis test with Dunn's multiple comparison correction (when Gaussian distribution was not assumed). Adjusted p values are reported. All differences were considered statistically significant at p < 0.05. Data are presented as mean ± standard deviation in bar graphs or a dot graph as appropriate. For correlation analysis between continuous variables, the Spearman correlation

coefficient (rs) was determined, where rs > 0.5 and p < 0.05 were considered high correlation. For correlation analysis involving ordinal variables, the Kendall's tau-b coefficient ($\tau$) was determined.

## Acknowledgements

We are grateful to Guangzhou Gene Denovo Biotechnology Co, Ltd, China, for assistance in RNA sequencing and original data analyses. The present work has received no external funding but only from the affiliated institution.

## Additional information

### Funding

No external funding was received for this work.

### Author contributions

Zhiyuan Wu, Lindi Liu, Formal analysis, Investigation, Methodology, Writing – original draft, Writing – review and editing; Chenchen He, Formal analysis, Methodology, Writing – original draft, Writing – review and editing; Lin Xiao, Resources, Methodology, Writing – review and editing; Duo Yun, Junliang Chen, Zhihao Liu, Wenjun Li, Methodology, Writing – review and editing; Qingjie Lv, Conceptualization, Resources, Supervision, Writing – review and editing; Xiaodong Tan, Conceptualization, Resources, Funding acquisition, Writing – review and editing

### Author ORCIDs

Zhiyuan Wu ⓘ https://orcid.org/0000-0002-2527-0635
Chenchen He ⓘ https://orcid.org/0000-0001-9082-2611
Xiaodong Tan ⓘ http://orcid.org/0000-0003-0862-1306

### Ethics
All protocols involving human subjects were approved by the Institutional Review Board (#2023PS953K), while written informed consent was exempted as only retrospective, de-identified clinical data and archival pathology specimens were used without jeopardizing the clinical treatment of the patients. All protocols for animal studies were approved by the Institutional Animal Care and Use Committee (#2022PS856K).

Reviewer #1 (Public review): https://doi.org/10.7554/eLife.102637.2.sa1
Reviewer #2 (Public review): https://doi.org/10.7554/eLife.102637.2.sa2
Author response https://doi.org/10.7554/eLife.102637.2.sa3

## Additional files

### Supplementary files
Supplementary file 1. Gene set enrichment analysis (GSEA) pathway analysis of colon tissues between wt and CD131-deficient mice.

Supplementary file 2. List of intersected differentially expressed genes (DEGs).

Supplementary file 3. Gene Ontology (GO) enrichment analysis of the intersected, up-regulated differentially expressed genes (DEGs).

Supplementary file 4. Gene Ontology (GO) enrichment analysis of the intersected, down-regulated differentially expressed genes (DEGs).

Supplementary file 5. Kyoto Encyclopedia of Genes and Genomes (KEGG) enrichment analysis of the intersected, up-regulated differentially expressed genes (DEGs).

Supplementary file 6. Kyoto Encyclopedia of Genes and Genomes (KEGG) enrichment analysis of the intersected, down-regulated differentially expressed genes (DEGs).

Supplementary file 7. Demographic and clinical data of ulcerative colitis patients. ESR: erythrocyte sedimentation rate; Fobt: fecal occult blood test; na: not available (data missing from clinical

registry).
MDAR checklist

## Data availability

The original RNA-seq data that support the findings of the present study have been deposited into CNGB Sequence Archive (CNSA) of China National GeneBank DataBase (CNGBdb) with accession number CNP0005868. Other data generated or analyzed in the present study have been deposited in Zenodo (https://doi.org/10.5281/zenodo.15760477). Data of human subjects have been made de-identified due to ethical considerations and authority regulations. All materials, including animals, cells, and reagents, used in the present study are commercially available. Data processing and analysis softwares include MS Excel, GraphPad Prism, IBM SPSS, Nikon NIS, FlowJo, ImageJ, as well as R software and packages, which are either commercially available or open source. The present study did not produce any new code or software for analyzing data.

The following datasets were generated:

| Author(s) | Year | Dataset title | Dataset URL | Database and Identifier |
|---|---|---|---|---|
| Zhiyuan W | 2024 | The effects of CD131 on experimental murine colitis | https://doi.org/10.26036/CNP0005868 | CNGBdb, 10.26036/CNP0005868 |
| Wu Z | 2025 | Data for: CD131 contributes to ulcerative colitis pathogenesis by promoting macrophage infiltration | https://doi.org/10.5281/zenodo.15760477 | Zenodo, 10.5281/zenodo.15760477 |

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
