## [Editor Report · eLife Assessment]

Ulcerative colitis (UC) is a chronic gut inflammatory condition affecting the colon in humans. This study uses human samples as well as a mouse model of colitis induced by a chemical, DSS, to investigate the role of an immune marker, CD131, in UC pathogenesis. The study, as presented, is **incomplete**, as experimental details are lacking, the statistical analyses are deficient, and there is not yet direct evidence for a CD131-mediated mechanism of gut inflammation.

---

## [Referee Report · Reviewer #1 (Public review)]

Summary:

This study investigates the role of CD131, a receptor subunit for GM-CSF and IL-3, in ulcerative colitis pathogenesis using a DSS-induced murine colitis model. By comparing wild-type and CD131-deficient mice, the authors demonstrate that CD131 contributes to DSS-induced colitis, working in concert with tissue-infiltrating macrophages.

Strengths:

The research shows that CD131's influence on macrophage and T cell chemotaxis is mediated by CCL4. The authors conclude by proposing a pro-inflammatory role for CD131 in murine colitis and suggest potential clinical relevance in human inflammatory bowel disease.

Weaknesses:

The statistical association between increased CD131 expression and clinical IBD was not observed in Table 1, indicating that the main results from animal experiments were not reproduced in human subjects. Additionally, due to the absence of experimental results regarding the downstream signaling pathways through CD131, it is difficult to infer the precise differentiated outcomes of this study. Furthermore, the effects of CD131 on immune cells other than macrophages were not presented, and the results specific to macrophage-selective CD131 were not shown. Therefore, I conclude that it is challenging to provide a detailed review as there is a lack of supporting evidence for the core arguments made in this paper.

---

## [Referee Report · Reviewer #2 (Public review)]

Summary:

This study investigates the potential role of CD131, a cytokine receptor subunit shared by GM-CSF and IL-3, in intestinal inflammation. Using heterozygous mice with an inactivating mutation on this gene, the study demonstrates ameliorated inflammation, associated with less infiltration of macrophages. Moreover, the depletion of macrophages prevented many of the inflammatory effects of DSS and made both WT and mutant mice equivalent in terms of inflammation severity. Correlative data showing increased CD131+ cells in tissues of patients with ulcerative colitis is also demonstrating, evidence for plausibility for these pathways in human disease.

Strengths:

The phenotype of mutant mice seems quite robust and the pathways proposed, GM-CSF signaling in macrophages with CCL4 as a downstream pathway, are all plausible and concordant with existing models. Many of the experiments included meaningful endpoints and were overall well performed.

Weaknesses:

(1) Experimental rigor was lacking in this manuscript, which provided limited or no details on the number of independent iterations that each experiment was done, the number of animals per group, the number of technical or biological replicates in each graph, etc.

(2) Details of animal model validation showing that this particular mutant allele results in a lack of CD131 protein expression were not shown. Moreover, since the paper uses heterozygous mice, it is critical to show that at the protein level, there is indeed reduced expression of CD131 in het mice compared to controls (many heterozygous states do not lead to appreciable protein depletion).

(3) Another major weakness is that the paper asserts a causal relationship between CD131 signaling and CCL4 production: the data shown indicates that the phenotypes of CCL4 deficiency (through Ab blockade) and CD131 partial deficiency (in het mice) are similar. However, this does not establish that CD131 signaling acts through CCL4.

(4) Lastly, while the paper claims that CD131 acts through macrophage recruitment, the evidence is circumstantial and not direct. DSS-induced acute colitis is largely mediated by macrophages, so any manipulation associated with less severe inflammation is accompanied by lesser macrophage infiltration in this model: this does not directly establish that CD131 acts directly on macrophages, which would require cell-specific knockout or complex cell reconstitution experiments.

---

## [Author Response]

**Our response to Reviewer #1:**

We appreciate the reviewer’s comments to clarify the strengths and weaknesses of our work. Whether the effect of GM-CSF/IL-3 on the bowel is pro-inflammatory or anti-inflammatory has been controversial. In the present study, we have shown that CD131 mediated a pro-inflammatory effect of GM-CSF on the intestine, which may have worked in synergy with tissue-infiltrating macrophages. While its down-stream signaling has been investigated back and forth, we did not put effort into it. Using macrophage-specific CD131-deficient animals is important to clarify the effects of macrophage-specific CD131 on bowel inflammation. Our present work is indeed incomplete, and we anticipate to work on it further in future research. Concerning the results on human subjects, it is indeed that results from animal experiments were not completely reproduced. We believe that CD131 does have an effect on ulcerative colitis; however, due to the use of biological agents (e.g. anti-TNFs), the need for surgery in the treatment of ulcerative colitis has dramatically decreased and we could not get enough samples to reach a more convincing statistical analysis. Twenty-nine patients shown in the present study were all that received surgical intervention at our center during the past decade, and more human subjects will be needed in future research, possibly from multi-center study.

**Our response to Reviewer #2:**

Many appreciations for the valuable reviewer’s comments and suggestions. We realized that the number of animals per group was not indicated in each figure; in order to clarify the experimental rigor, we have deposited data used to generate the results of the present study in Dryad. Concerning the heterozygous CD131 knock-out animals, we think that others have used the homozygous mice in their studies; however, we observed premature deaths in those animals and we could not get any single homozygous mouse. We could not tell the exact reason, but we did observe robust phenotypes in these heterozygous mice. We do realize that our present work is incomplete, and more experiments need to be done to establish a causal relationship between CD131 and down-stream effects. We anticipate to use macrophage-specific homozygous CD131-deficient mice in our future research, which we believe will produce more meaningful and convincing results.